# Machine learning and causal inference applied to the gut metagenome-metabolome axis reveals a link between neonatal jaundice and autism spectrum disorder

Xianhong Chen,[1,2,3] Cheng Chen,[2] Xiucai Lan,[4] Xueli Zhang,[5] Tingting Li,[2] Peng Zhang,[6] Guoqiang Cheng,[6] Wei Zhou,[7] Zhangxing Wang,[5] Yingmei Xie,[2] Shujuan Zeng,[3] Wenhao Zhou,[7] Mingbang Wang[2]

**ABSTRACT**  Neonatal jaundice (NJ) might increase the risk of autism spectrum disorder (ASD) in children. This study examined whether alterations in the gut microbiota could explain the link between NJ and ASD. We analyzed three cohorts: NJ cohort 1 comprised 68 neonates with NJ and 68 healthy controls (HCs); NJ cohort 2 included 56 infants with NJ and 14 HCs; and the ASD cohort consisted of 43 children with ASD and 31 typically developing children. Fecal samples were collected aseptically. We performed 16S rRNA sequencing (NJ cohort 1), liquid chromatography with tandem mass spectrometry metabolomics (NJ cohort 1 and ASD cohort), and shotgun metagenomics (NJ cohort 2 and ASD cohort). We characterized the gut DNA virome, quantified bile acid metabolism genes, and integrated multi-omics data using causal mediation and machine learning causal inference. Both NJ and ASD were associated with increased diversity of bile acid metabolism genes, suggesting biomarker potential. The gut DNA virome was also identified as a potential biomarker. Causal mediation analysis showed that the gut DNA virome influences bile acid metabolism genes in both conditions. Using machine learning-based causal modeling, we further found that gut *human betaherpesviruses* and *human mastadenoviruses* contribute to NJ and ASD, respectively, mediated by gut bile acid-metabolizing bacteria. These findings suggest that perturbations in the virome and bile acid-metabolizing bacteria may explain the link between NJ and ASD. Our results indicate that NJ and ASD are associated with bile acid metabolism alterations, which are also influenced by the gut DNA virome. Dysbiosis of the gut DNA virome and bile acid-metabolizing bacteria may mechanistically link NJ and ASD.

**IMPORTANCE**  Human epidemiological studies have established an association between perinatal pathogenic infections and autism spectrum disorder (ASD), and the gut microbiota plays an extremely important role in this relationship. Neonatal jaundice (NJ) may increase the risk of ASD in children. However, it remains unclear whether alterations in the gut microbiota affect the association between NJ and ASD. Both NJ and ASD are linked to altered gut bile acid metabolism and significantly elevated gene diversity among bile acid metabolism enzymes, and these relationships are influenced by the gut virome. Gut human betaherpesviruses and human mastadenoviruses influence the development of NJ and ASD, respectively, by influencing the abundance of gut bile acid-metabolizing microbes. Alterations of the gut virome and bile acid-metabolizing bacteria appear to explain the link between NJ and ASD. There is a lack of effective treatment options for ASD. We found that both NJ and ASD are linked to altered bile acid metabolism. Gaining a comprehensive understanding of the role of the bile acid-gut microbiota axis in the pathogenesis of NJ and ASD, as well as regulating this axis, may be crucial for developing novel preventive and therapeutic strategies for ASD.

**Peer Reviewer** Xingyin Liu, Nanjing Medical University, NanJing, China

Address correspondence to Shujuan Zeng, zengshujuansz@163.com, Wenhao Zhou, zhouwenhao@fudan.edu.cn, or Mingbang Wang, mingbang.wang.bgi@qq.com.

Xianhong Chen, Cheng Chen, and Xiucai Lan contributed equally to this article. The order of the first authors was determined through mutual agreement based on their complementary contributions to the study design, execution, and manuscript preparation.

The authors declare no conflict of interest.

See the funding table on p. 13.

**KEYWORDS** gut microbiota, virome, bile acid metabolism, autism spectrum disorder, neonatal jaundice

Human epidemiological evidence suggests that prenatal maternal infection by rubella, herpes simplex virus type 2, cytomegalovirus, or other pathogens and the resulting immune activation might influence the pathogenesis of neuropsychiatric disorders such as autism spectrum disorder (ASD) (1–6). The intraperitoneal administration of lipopolysaccharide (mimicking bacterial infection) or poly I:C (mimicking viral infection) to mice at day 12.5 of conception induces maternal immune activation (MIA); offspring born to mothers with MIA have been used as a classical animal model to understand the pathogenesis of ASD because they often exhibit ASD-like behavioral abnormalities (7–9). In support of the epidemiological data, studies based on animal models of MIA indicated that the maternal gut microbiota promotes neurodevelopmental abnormalities in mouse offspring (10), and human clinical studies also revealed that fecal microbiota transplantation (FMT) alters the gut microbiota composition of children with ASD while improving their behavioral symptoms (11). In conclusion, human epidemiological studies have established a link between perinatal pathogenic infections and ASD, and studies have indicated that alterations of the gut microbiota play an extremely important role in this association.

Neonatal jaundice (NJ) can increase the risk of ASD in children (12–14). NJ is a common neonatal complication, with its prevalence reaching 80% in preterm infants (15, 16), and preterm birth is associated with an increased risk of ASD (17, 18). NJ can increase the risk of ASD in children, and alterations of the gut microbiota link perinatal/infant risk factors with behavioral abnormalities. Disturbances of the gut microbiota early in life can affect brain development and behavior in offspring. Using a cohort of 213 mothers and 215 children, Dawson et al. found that alpha diversity in the maternal fecal microbiota during late gestation predicted internalizing behavior in children at 2 years of age, and taxa from the butyrate-producing families Trichophyceae and Coccidae were more abundant in the mothers of children with normal behavior (19). A healthy prenatal diet indirectly reduces internalizing behavior in children by increasing the alpha diversity of the maternal gut microbiota (19). There is a strong association between the composition of the gut microbiota in infancy and subsequent behavioral problems. By analyzing the gut microbiota at 1, 6, and 12 months of age and measuring behavioral outcomes at 2 years of age in 201 children, Loughman et al. found a significant association between the decreased abundance of *Prevotella* spp. in fecal samples collected at 12 months of age and increased behavioral problems at 2 years of age (20).

The gut microbiota is involved in the metabolism of bile acids, and the composition of the gut microbiota is influenced by bile acids themselves (21). A comprehensive understanding of the gut bacteria involved in the metabolism of bile acids is necessary. Bile acids are cholesterol-derived steroid molecules that play important roles in energy homeostasis, host metabolism, and the maintenance of innate immunity through G protein-coupled receptors and/or nuclear receptors (22). There is a bidirectional interaction between the gut microbiota and bile acids. On the one hand, bile acids reshape the gut microbiota by promoting the growth of bile acid-metabolizing bacteria and inhibiting the growth of other bile-sensitive bacteria, and on the other hand, the gut microbiota can influence bile acid metabolism and the composition of the bile acid pool by producing a variety of enzymes, such as bile salt hydrolases (BSHs) and hydroxysteroid dehydrogenases (HSDHs), which modify primary bile acids to secondary bile acids and further degrade them via other enzymatic mechanisms, thus helping to maintain cholesterol homeostasis (23).

The mechanisms by which the human gut DNA virome is established early in life and by which it interacts with the microbiota and metabolome, thereby playing a role in human disease, are unclear. Multiple enterovirus communities that infect both microbial and animal hosts collectively constitute the human gut DNA virome, and recent advances in human gut metagenomic sequencing and analysis revealed the complexity

of the human gut DNA virome and facilitated the discovery of new viruses (24, 25). We previously used multi-omic tools such as metagenome–metabolome bioinformatic analysis to capture key bacteria involved in critical glutamate metabolism in the gut microbiota of patients with ASD and revealed an increase in the abundance of the bile acid-metabolizing bacterium *Eggerthella lenta* and its association with glutamate metabolism (26). Newly developed causal inference methods have also accelerated the use of multi-omic techniques to unravel the underlying pathogenesis of complex diseases (27–29). In the present study, we explored the landscape of the gut DNA virome, bile acid-metabolizing bacteria, and the metabolome in patients with NJ and ASD through a multi-omics approach and used machine learning causal inference methods to obtain a comprehensive understanding of the composition of the gut microbiota and metabolites that influence the clinical symptoms of NJ and ASD.

## MATERIALS AND METHODS

### Participant recruitment and sample collection

In total, three cohorts were included in this study. NJ cohort 1 included 68 patients with NJ and 68 matched healthy controls (HCs). NJ cohort 2 included 56 infants with NJ and 14 HCs. The ASD cohort included 43 children with ASD and 31 typically developing (TD) children. The inclusion and exclusion criteria for the three cohorts are detailed in the Supplemental methods. Researchers wearing medical sterile masks and sterile gloves collected fresh fecal samples within 1 h of participants' defecation using disinfected disposable fecal collection tubes to prevent contamination. All samples were assigned unique identifiers to ensure anonymity. Samples were processed within 2 h of collection, aliquoted into 3–5 g portions per tube, and stored at −80°C. Written informed consent was obtained from the parents of each participant, and the study was approved by the Longgang District Maternity and Child Healthcare Hospital (approval no. LGFYKYXMLL-2024-100). Details of the cohorts are provided in Tables S1 to S3. The overall design of this project is presented in Fig. 1.

### 16S rRNA gene sequencing

In NJ cohort 1, 16S rRNA gene sequencing was used to detect the composition of the gut microbiome with reference to previously published studies (30–33), as detailed in the Supplemental material.

### Gut metabolome analysis

In NJ cohort 1 and the ASD cohort, in reference to our previous studies (26, 34, 35), we analyzed the gut metabolome using liquid chromatography with tandem mass spectrometry (LC-MS/MS). Briefly, approximately 200 mg of fresh fecal sample per subject was used for metabolite extraction. The detailed procedures for sample preparation, LC-MS/MS instrumentation, and data processing are provided in the Supplemental material.

### Gut metagenomic analysis

In NJ cohort 2 and the ASD cohort, referring to our previous studies (26, 36–38), we performed shotgun metagenomic sequencing, as detailed in the Supplemental material.

### Gut bile acid metabolism genes

In NJ cohort 2 and the ASD cohort, metagenomic data sets were used to systematically identify gut bacteria carrying BSH and HSDH genes, as detailed in the Supplemental material.

### Gut DNA virome

In NJ cohort 2 and the ASD cohort, metagenomic data sets, together with PathoScope software (version 2.0) (39, 40), were used to identify viral sequences from metagenomic

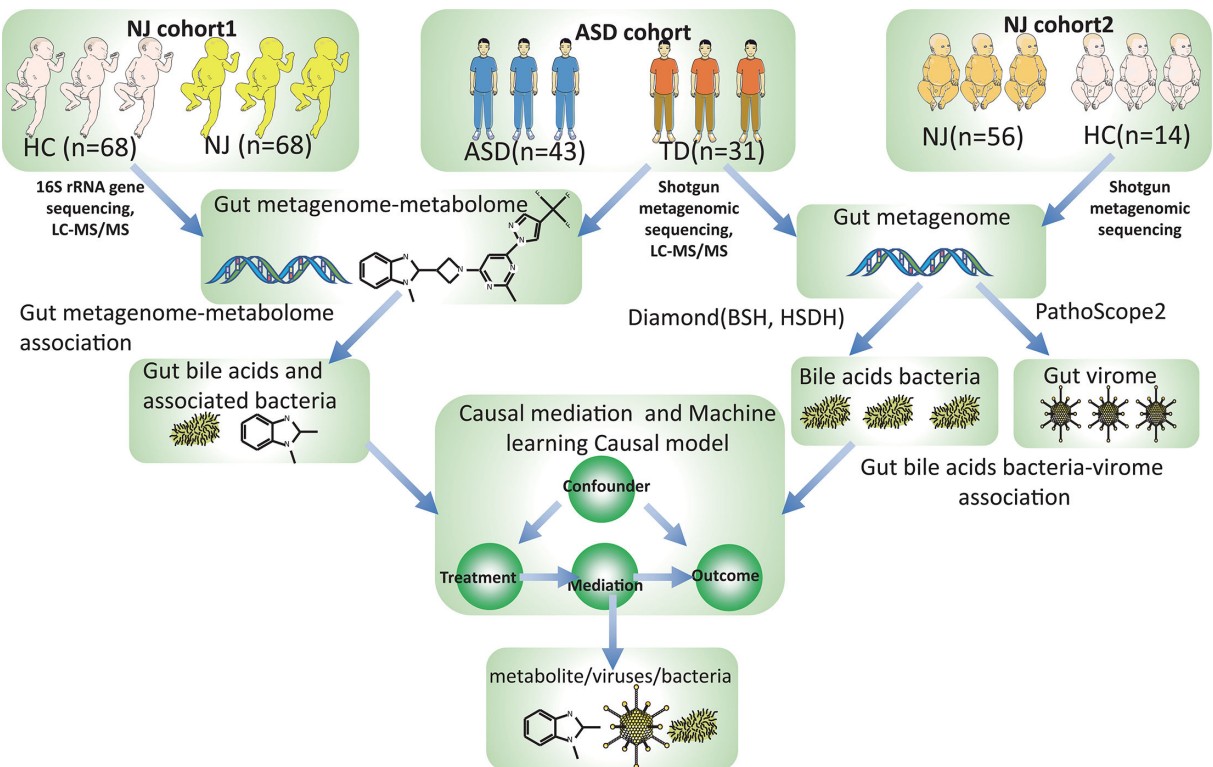

**FIG 1** Study flowchart. This study included three cohorts, NJ cohort 1 (68 patients and 68 HCs), NJ cohort 2 (56 patients and 14 HCs), and the ASD cohort (43 children with ASD and 31 TD children). The overall design of this study was as follows. (i) Identification of metabolome alterations in NJ and ASD: liquid chromatography with tandem mass spectrometry (LC-MS/MS) was used to analyze metabolomes in NJ cohort 1 and the ASD cohort. In NJ cohort 1, 16S rRNA gene sequencing was used to profile the gut microbiome. For the ASD cohort, shotgun metagenomic sequencing was employed for profiling. Microbiome-metabolome association analysis identified bacteria and metabolites linked to NJ and ASD. (ii) Association of bile acid metabolism genes with the gut DNA virome: Shotgun metagenomic data from NJ cohort 2 and the ASD cohort were analyzed. The gut DNA virome was characterized using PathoScope 2, and bile acid metabolism genes (BSH and HSDH) were quantified with Diamond. Associations among viruses, bile acid metabolism genes, NJ, and ASD were examined through virome-metagenome integration. (iii) Linking gut DNA virome and bile acid metabolism to NJ and ASD: Causal mediation analysis evaluated the effects of the gut DNA virome and bile acid metabolism genes on NJ/ASD and clinical indicators. Then, we constructed a causal model and evaluated the robustness of the causal model using a machine learning causal inference method.

data, as detailed in the Supplemental material. It is important to note that the shotgun metagenomic sequencing approach employed in this study captures only DNA sequences. Consequently, our analysis of the viral community is confined to the DNA virome, and RNA viruses were not detected.

## Procrustes analysis

Procrustes analysis was used to assess the concordance of significant markers of differences in multi-omics data using the R package vegan as described by Zhao et al. (41) and as detailed in the Supplemental material.

## Causal mediation analysis

Causal mediation analysis was used to assess the causal relationship of the independent variable with the dependent variable based on the mediating variable using the R package mediation (version 4.5.0), as detailed in the Supplemental material.

## Causal inference based on machine learning

Causal inference based on machine learning was performed using Microsoft's DoWhy library (https://github.com/microsoft/dowhy) and the EconML library (https://

github.com/econml/), as detailed to in the software manual, and the protocol was described in previous studies (42–45) and the Supplemental material.

## Other statistical analyses

To evaluate whether differences in gut microbiota composition, metabolites, virome, or bile acid metabolism genes could be used to distinguish patients with NJ from HCs or patients with ASD from TD children, we employed partial least squares discriminant analysis (PLS-DA). This analysis was conducted using the plsda() function from the R package mixOmics. To examine the correlations of the gut microbiota composition, metabolites, virome, or bile acid metabolism genes that differed significantly between the groups with clinical performance, logistic regression models were fitted using the lm() function in R. The resulting *P*-values and coefficients of determination were extracted via the summary() function. Additionally, boxplots and scatterplots were generated using the beeswarm R package to visualize group differences, with statistical significance determined by the Wilcoxon rank-sum test (wilcox.test() function).

## RESULTS

### Children with NJ or ASD exhibit alterations in gut bile acid metabolism

Through the integration of multi-omics data and machine learning-based causal inference in our previous studies (42), we identified significant contributions of gut bacteria and bile acid metabolites in NJ, allowing for the prediction of NJ risk. To investigate the association between gut metabolome alterations and NJ or ASD, LC-MS/MS was employed to characterize the metabolomic profiles of both NJ cohort 1 and the ASD cohort. In NJ cohort 1, the gut microbiome was profiled using 16S rRNA gene sequencing, while shotgun metagenomic sequencing was applied to analyze the gut microbiome in the ASD cohort.

We found that infants with NJ exhibited significantly higher gut bile acid metabolism ($P < 0.001$; Fig. S1A) and a marked reduction in the levels of the gut bile acid metabolite tauroursodeoxycholic acid compared to HCs ($P < 0.001$; Fig. S1B). In addition, tauroursodeoxycholic acid levels were negatively correlated with serum total bilirubin (TBIL) levels (Fig. S1C). Notably, patients with ASD also exhibited altered gut bile acid metabolism, demonstrating significantly elevated levels of the bile acid metabolites chenodeoxycholic acid-3-sulfate ($P < 0.05$; Fig. S1D) and taurocholic acid ($P < 0.05$; Fig. S1E) compared to TD children. Furthermore, chenodeoxycholic acid-3-sulfate levels showed a positive correlation with IgA levels (Fig. S1F).

### Altered gut bile acid metabolism associated with the gut DNA virome composition

To elucidate the role of gut microbiota in bile acid metabolism in patients with NJ and ASD, we conducted shotgun metagenomic sequencing of fecal samples from both NJ cohort 2 and ASD cohort. The key enzymes involved in bile acid metabolism in intestinal bacteria are BSH and HSDH. BSH hydrolyzes bound bile salts to free bile acids and glycine/taurine, and HSDH further oxidizes free bile acids to form secondary bile acids, thereby regulating bacterial populations and maintaining gut microbiota balance (46, 47). We systematically quantified the abundance of BSH and HSDH genes in the metagenomic data sets to characterize the gut bile acid-metabolizing bacterial communities in these patient groups. Furthermore, given the potential regulatory role of gut viruses in shaping bacterial communities, we performed comprehensive virome profiling to identify bacteriophages and pathogenic viruses that may influence bile acid-metabolizing bacteria.

Through metagenomic association analysis, we identified specific BSH and HSDH genes associated with each disease. The bile acid metabolism genes of NJ include WP_035763572.1, 2BJF_A, WP_011590332.1, etc. Genes associated with ASD include WP_182424780.1, AAO76366.1, WP_057826614.1, RHH16704.1, etc. The complete list

of BSH and HSDH genes associated with NJ and ASD is provided in Tables S4 and S5, respectively. Based on PLS-DA, we found that the NJ and HC groups could be distinguished to some extent based on the abundance of gut bile acid metabolism genes (Fig. 2A), and these groups could also be clearly distinguished using the gut DNA virome composition (Fig. 2B). Furthermore, we assessed the consistency of significantly altered gut bile acid metabolism genes and changes in the gut DNA virome composition based on Procrustes analysis, and the results revealed the potential consistency of differentially expressed gut bile acid metabolism genes and differences of the gut DNA virome composition between the NC and HC groups ($P < 0.001$; Fig. 2C).

Similar to the results in NJ cohort 2, based on PLSDA analysis, we found that the abundance of gut bile acid metabolism genes could somewhat distinguish children with ASD from TD children (Fig. 2D), and the gut DNA virome composition could clearly distinguish children with ASD from TD children (Fig. 2E). Furthermore, we also evaluated the consistency of differentially expressed gut bile acid metabolism genes and differences of the gut DNA virome composition based on Procrustes analysis, and the results demonstrated the potential consistency of differentially expressed gut bile acid metabolism genes and differences of the gut DNA virome composition between the ASD and TD groups ($P < 0.001$; Fig. 2F).

To investigate potential associations between gut viruses and key components of bile acid metabolism, we examined correlations with both bile acid-metabolizing bacteria and related functional genes. Our analysis revealed several specific virus-microbe relationships in the NJ cohort 2 (Fig. S3A and B). Most notably, the abundance of *Bifidobacteriales* was inversely associated with human betaherpesvirus levels ($P < 0.001$, $R^2 = 0.22$), and similarly, *Bifidobacterium* showed a negative correlation with *Escherichia* Stx1 phage ($P < 0.001$, $R^2 = 0.22$). In contrast, we found a positive correlation between

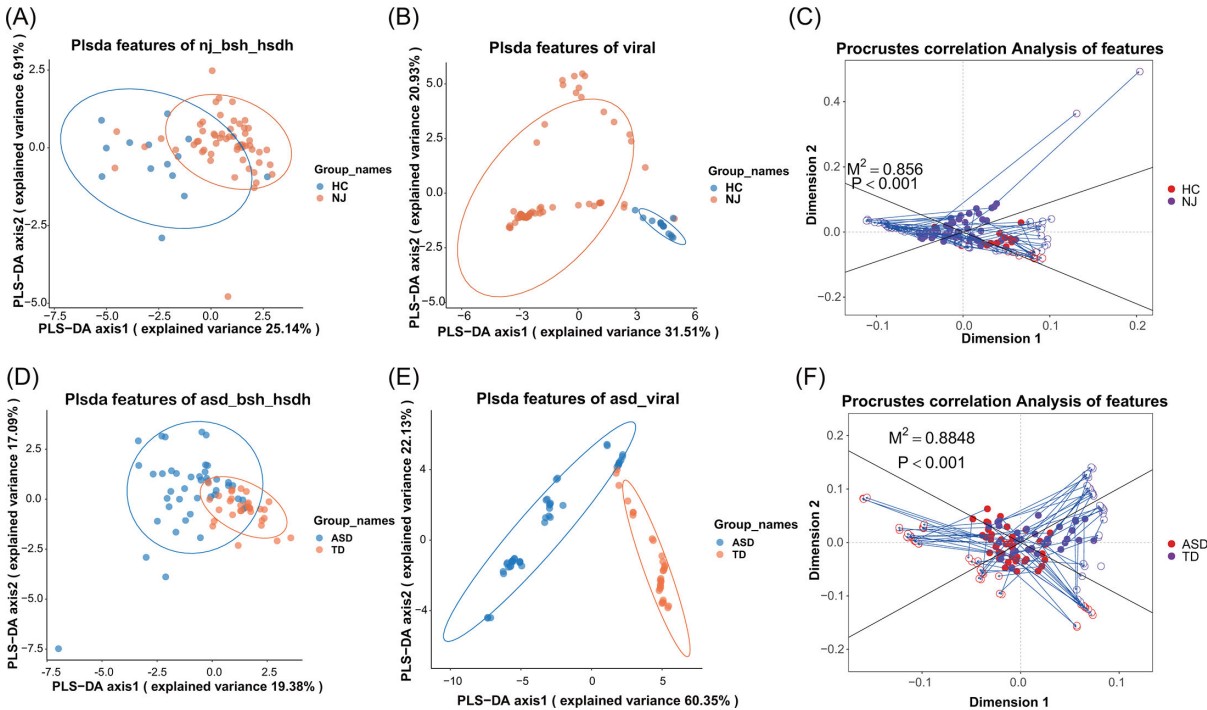

FIG 2 Altered gut bile acid metabolism genes associated with the DNA viral composition. (A–C) Results from NJ patients and HCs. (A) PLS-DA revealed that the NJ and HC groups could be distinguished to some extent based on the abundance of gut bile acid metabolism genes. (B) PLS-DA revealed that the NJ and HC groups could be distinguished to some extent based on the DNA viral composition. (C) Procrustes analysis indicating significant concordance between DNA viral composition and bile acid metabolism gene abundance in NJ and HC groups ($P < 0.001$). Panels D–F present the results of patients with ASD and TD children. (D) PLS-DA plot distinguishing ASD and TD groups based on gut bile acid metabolism gene abundance. (E) PLS-DA plot clearly separating ASD and TD groups based on DNA viral composition. (F) Procrustes analysis demonstrating significant concordance between DNA viral composition and bile acid metabolism gene abundance in ASD and TD groups ($P < 0.001$).

*Mycobacterium* phage and the bile acid metabolism gene WP_17534087 ($P < 0.001$, $R^2 = 0.27$). Within the ASD cohort, *human mastadenovirus* levels were positively associated with the bile acid metabolism gene WP_002565860_1 ($P < 0.001$, $R^2 = 0.28$; Fig. S3D). Conversely, this same gene showed negative correlations with both *Enterobacterial* phage ($P < 0.001$, $R^2 = 0.24$; Fig. S3eE) and *Pandoravirus* dulcis ($P < 0.001$, $R^2 = 0.32$; Fig. S3F). These findings reinforce the potential functional links between specific viral elements and microbial (including gene-level) features in both NJ and ASD.

To further evaluate the clinical significance of bile acid metabolism genes, we examined their correlations with clinical phenotypes. The results indicated that in neonates with NJ, the abundance of gut bile acid metabolism genes (including EEA86306_1, EEG48917_1, and WP_ 082174150_1) was positively correlated with serum direct bilirubin (DBIL) levels (Fig. S2A through C). Similarly, among children diagnosed with ASD, a positive association was observed between the abundance of intestinal bile acid metabolism genes (such as WP_002565680.1, WP_002587930.1, and WP_007662998.1) and serum IgA levels (Fig. S2D through F).

## Gut DNA virome composition affects NJ and ASD through bile acid metabolism genes

To further investigate whether the gut DNA virome and bile acid metabolism genes share mechanistic features that could partly explain the parallel findings in NJ and ASD, we performed causal mediation analysis to assess the effects of key viral constituents and bile acid metabolism genes on the development and core clinical features of each condition separately. In NJ cohort 1, the analysis revealed that the gut DNA virome can exert indirect physiological effects through specific bile acid metabolism genes, particularly 2HF0 and Q9KK62, while also exerting direct influences via genes such as AAC45411_2, EEA86306_1, and WP (Fig. 3A). Moreover, the virome composition also affected important clinical indicators of NJ, specifically serum DBIL and TBIL, through bile acid metabolism genes (Fig. 3B and C, detailed results in Table S6).

Similarly, in the ASD cohort, causal mediation analysis indicated that the composition of the enterovirus group could affect ASD through bile acid metabolism genes. Specifically, *Human mastadenovirus* was found to directly affect ASD via genes, including 6UFY_A, AAO76366_1, WP_002587930_1, and WP_007662998_1, while exerting indirect effects through AAB61150_1 (Fig. 3D). Furthermore, enteroviruses were shown to impact gut bacterial composition, particularly affecting species like *Acidaminococcus fermentans*, *Eggerthella*, *Bifidobacterium animalis*, and members of the *Erysipelotrichaceae* family (Fig. 3E). Additionally, the virome composition was found to influence gut bilirubin levels through its interaction with bile acid metabolism genes (Fig. 3F), as detailed in Table S7.

Finally, we constructed a causal model and evaluated the robustness of the causal model using a machine learning causal inference method. In NJ cohort 2, gut abundance of human betaherpesviruses was significantly elevated in the NJ group compared to HCs ($P < 0.01$; Fig. 4A) and was positively correlated with serum DBIL levels ($P = 0.0002$; Fig. 4B). Machine learning causal inference analysis suggested a potential causal effect of gut human betaherpesviruses on NJ, mediated by bacteria harboring bile acid metabolism genes (2HF0_A and 2HF0_B), with breastfeeding and the delivery mode identified as potential confounders (Fig. 4C).

Similar to the results in NJ cohort 2, analyses in the ASD cohort revealed significantly higher gut abundance of human mastadenoviruses in ASD children compared to TD controls ($P < 0.001$; Fig. 4D), which correlated positively with intestinal IgA content ($P < 0.001$; Fig. 4E). The machine learning-based causal model indicated a potential effect of human mastadenoviruses on the development of ASD, mediated by bile acid-metabolizing bacteria (carrying RHH16704_1 and 6UFY_A genes) and *Eggerthella* spp., while gastrointestinal problems and gender were identified as potential confounders (Fig. 4F). Through the summary and analysis of indicators such as gut bacteria, DNA viruses, bile acid-metabolizing bacteria, differentially expressed genes, group B *Streptococcus* (GBS) genes, and metabolites, we also found that a total of 70 gut markers were significantly

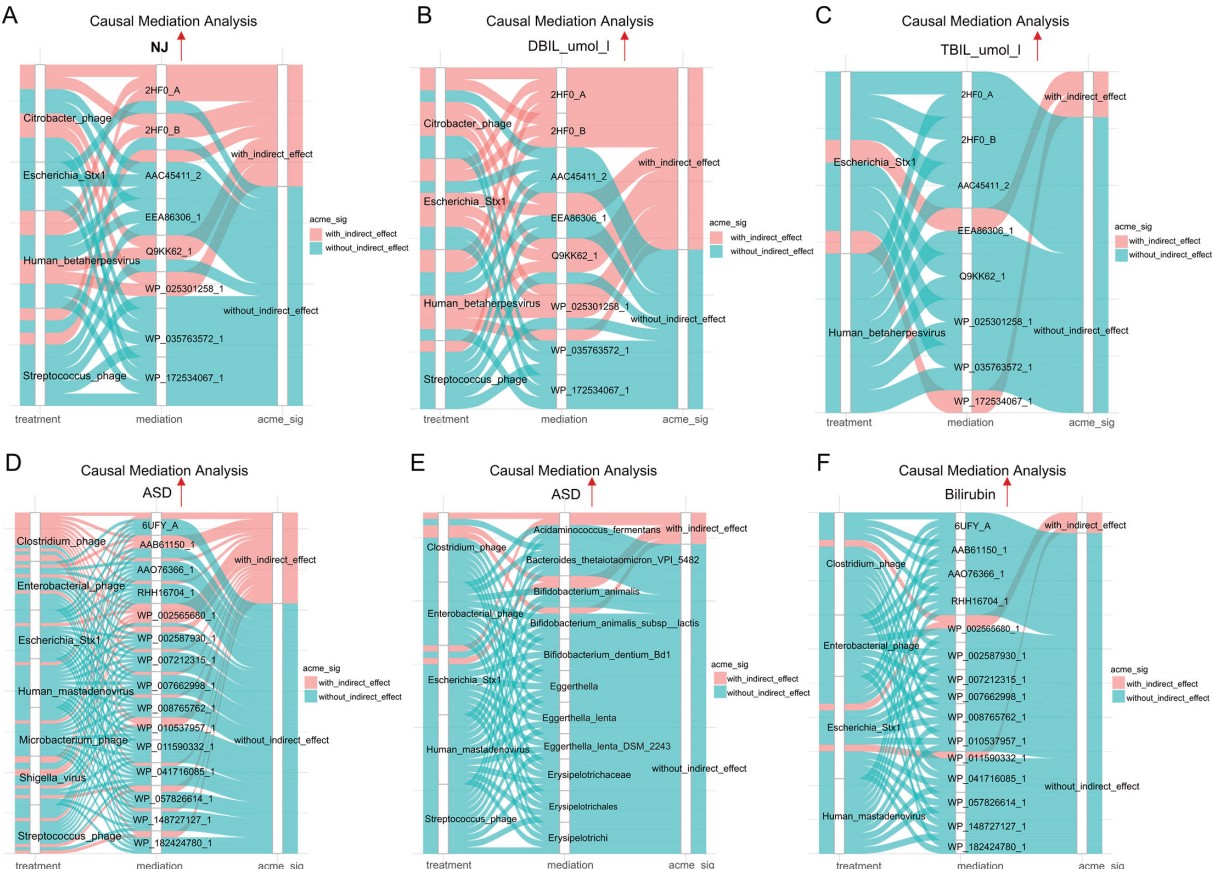

**FIG 3** Causal mediation analysis revealed enteroviruses' influence on NJ and ASD through bile acid metabolism genes. Results are visualized using alluvial diagrams. In each subplot, the three horizontal columns represent the causal effect of the treatment variable, the mediating variable, and the outcome variable mediated through the intermediate variable. Red indicates an indirect causal effect; blue indicates a direct causal effect. (A–C) Causal mediation analysis revealed that the DNA viral composition affects the development of NJ (A), DBIL levels (B), and TBIL levels (C) through bile acid metabolism genes. (D–F) Causal mediation analysis demonstrated that the DNA viral composition affects the development of ASD (D and E) and gut bilirubin levels (F) via bile acid metabolism genes. Model stability was evaluated via causal mediation analysis with 999 bootstrap replications to generate 95% CIs and $P$-values for the effects. A significant average causal mediation effect (ACME) ($P < 0.05$) confirms a stable causal mediation effect. The upward arrows in the figure indicate that these pathways lead to an increased risk of ASD and NJ or an increase in related indicators.

elevated in both the ASD and NJ groups. These included 66 gut bacteria, 2 gut bile acid metabolism genes, 1 GBS gene, and 1 gut metabolite. Meanwhile, 33 gut markers were significantly reduced in the disease groups, comprising 31 bacteria, 1 GBS gene, and 1 gut bile acid metabolism gene, as detailed in Table S8.

## DISCUSSION

Accumulating evidence underscores the critical role of the gut microbiota in human health, particularly through its involvement in bile acid metabolism (48–52). The neonatal gut microbiome is underdeveloped and highly vulnerable to environmental influences, such as medical care settings. Notably, neonates with NJ are often transferred to the neonatal intensive care unit, which has been identified as an environmental risk factor for ASD (53). This underscores the importance of the early-life gut microenvironment in neurodevelopment. Alterations in gut microbiota-mediated bile acid metabolism have been strongly associated with various liver diseases. For instance, Schneider et al. found that the gut microbiota and bile acid composition were altered in patients with primary sclerosing cholangitis and that gut microbiota-mediated negative feedback control of BA synthesis led to increased hepatic bile acid concentrations and disrupted bile duct barrier function (54). New evidence suggests that disturbances of gut

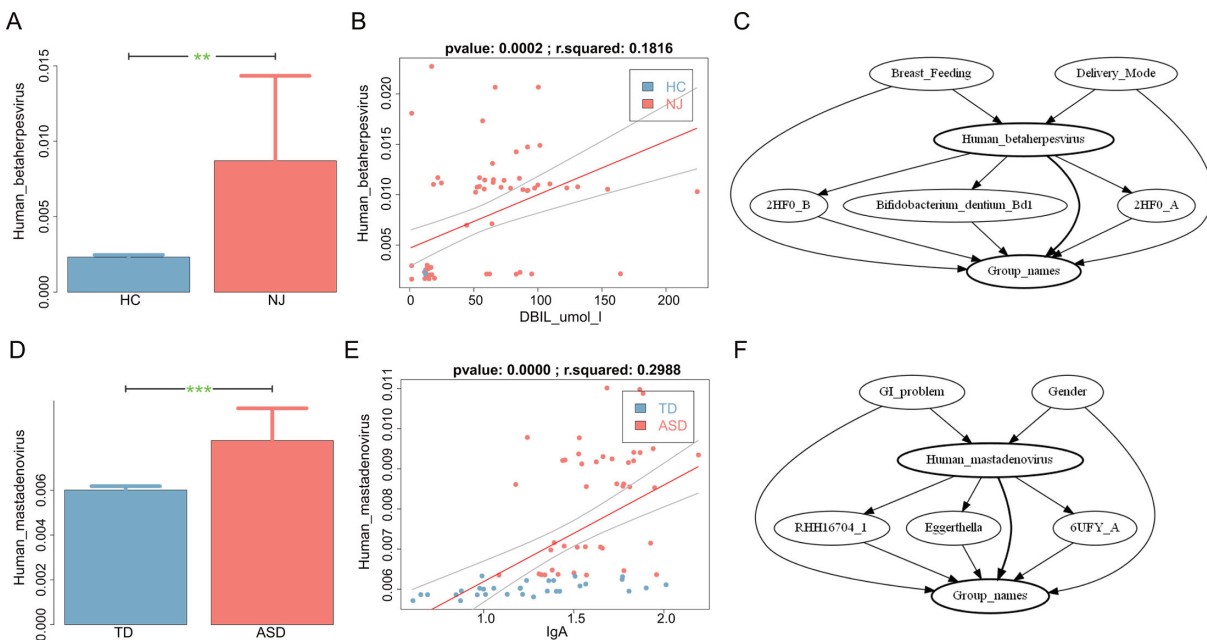

**FIG 4** Machine learning-based causal inference suggests gut DNA virome influences ASD and NJ through bile acid-metabolizing bacteria. (A–C) Gut *human betaherpesviruses* contribute to NJ development via bile acid-metabolizing microbes. (A) *Human betaherpesviruses* were more abundant in the NJ group than in the HC group ($P < 0.01$). (B) The abundance of human betaherpesviruses was significantly and positively correlated with serum DBIL levels. (C) Human betaherpesviruses have a potential causal effect on the development of NJ, and bile acid-metabolizing microbes (i.e., bacteria rich in the 2HF0_A and 2HF0_B genes as mediators), breastfeeding, and the delivery mode were identified as potential confounders. (D–F) Gut human mastadenoviruses affect the development of ASD by influencing gut microbes involved in bile acid metabolism. (D) Human mastadenoviruses were more abundant in ASD children than in TD controls ($P < 0.001$). (E) Significant positive correlation between the intestinal abundance of human mastadenoviruses and intestinal IgA content. (F) Machine learning causal inference methods identified a potential causal effect of human mastadenoviruses on the development of ASD, and bile acid-metabolizing microbes (i.e., bacteria rich in the RHH16704_1 and 6UFY_A genes) and *Eggerthella* spp. as mediators, gastrointestinal problems, and gender were identified as potential confounders. *$P < 0.05$, **$P < 0.01$, and ***$P < 0.001$.

microbiota-mediated bile acid synthesis play an important role in neurological disorders. It is known that bile acids can cross the blood-brain barrier and act directly on their receptors in the brain or more indirectly by activating intestinal receptors, leading to the release of signals such as fibroblast growth factor and glucagon-like peptide 1, thereby affecting neuronal activity in multiple brain regions or the vagus nerve (55–57). Proper regulation of bile acid homeostasis is essential for normal brain function; defects in these pathways are linked to neuropathological outcomes in both mice and humans, including demyelination, motor deficits, neuroinflammation, seizures, and cognitive impairment (58). In the present study, we observed that patients with NJ or ASD exhibited altered gut bile acid metabolism along with significantly increased diversity of gut bile acid metabolism genes, suggesting that these genes may serve as potential biomarkers for both conditions (see Fig. 5).

Gut microbiota-mediated Th17/interleukin-17A (IL-17A) abnormalities play extremely important roles in ASD (59, 60). Previous studies in mice demonstrated that the maternal IL-17A signaling can promote autism-like behavior in offspring in a gut microbiota-dependent manner (10, 61). Bile acids act as important immunomodulators that regulate a range of immune cell functions by activating receptors such as TGR5 and FXR (62, 63). Recent studies indicated that gut microbiota-derived bile acid metabolites can act as ligands for the key transcription factor retinoid-related orphan receptor-γt (RORγt) to directly regulate the differentiation of Th17 cells. For example, Huang et al. found that the bile acid metabolite 3-oxoLCA inhibited Th17 cell differentiation by directly binding RORγt, and the administration of 3-oxoLCA to mice reduced Th17 cell differentiation in the gut lamina propria (64). In our study, we identified a positive correlation between the abundance of specific bile acid metabolism genes (WP_002565680.1, WP_002587930.1,

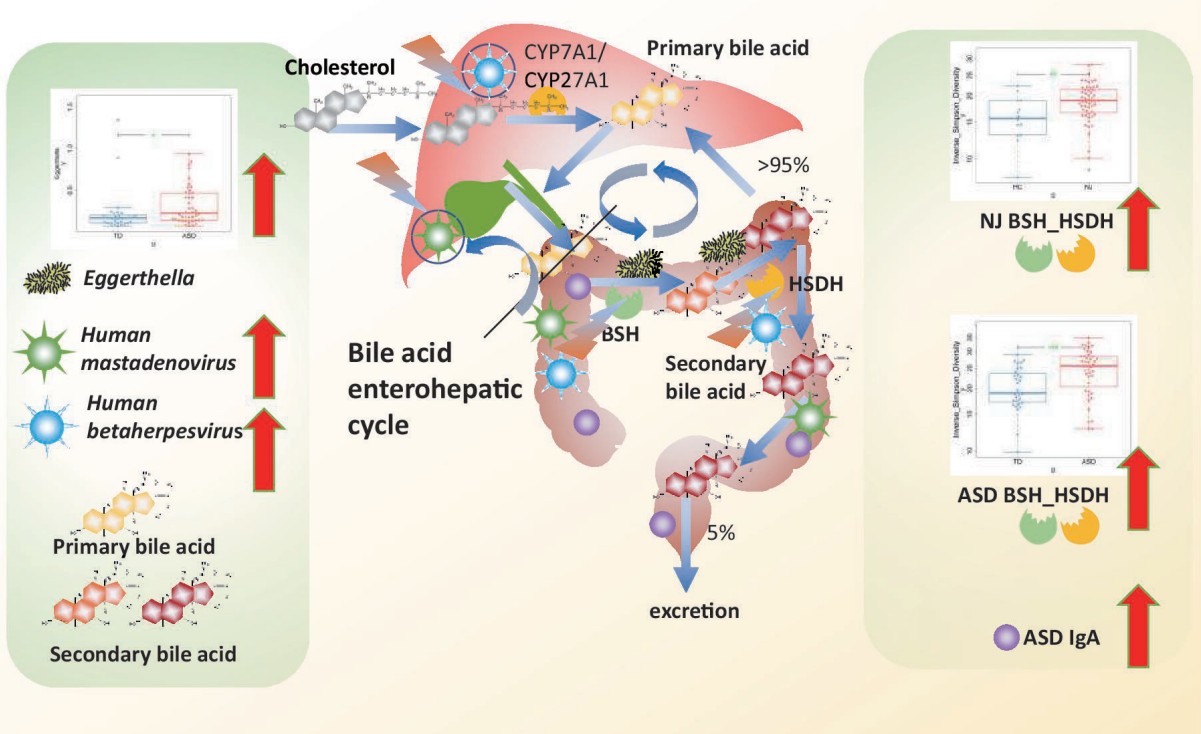

**FIG 5** Potential mechanisms by which alterations in the gut DNA virome and bile acid-metabolizing bacteria contribute to the development of NJ and ASD. Cholesterol is metabolized in the liver to synthesize primary bile acids, which are secreted into the intestine via the gallbladder. In the gut, bacterial BSH deconjugates primary bile acids into free bile acids, which are then transformed into secondary bile acids through oxidation catalyzed by bacterial HSDH. The majority (>95%) of intestinal bile acids are reabsorbed and returned to the liver via enterohepatic circulation. Alterations in the gut microbiota can disrupt this process, affecting bile acid homeostasis. Patients with NJ and ASD showed increased diversity of bile acid-metabolizing bacteria. Those with ASD also exhibited significant increases in the abundance of bile acid-metabolizing bacteria (*Eggerthella* spp.) and IgA levels. Gut bile acid metabolism was influenced by human mastadenoviruses and human betaherpesviruses. We propose that virome-induced changes in bile acid-metabolizing bacteria may contribute to the link between NJ and ASD.

and WP_007662998.1) and IgA levels in children with ASD. WP_002587930.1 and WP_002565680.1 encode choloylglycine hydrolase, a putative BSH gene potentially involved in bile acid deconjugation, while WP_007662998.1 encodes an short-chain dehydrogenase/reductase (SDR) family oxidoreductase, likely participating in bile acid oxidation. Additionally, levels of the gut bile acid metabolite chenodeoxycholic acid-3-sulfate were significantly elevated in children with ASD and correlated positively with IgA levels, supporting a potential interaction between bile acid metabolism and gut immune responses in ASD. Similarly, as observed in ASD, patients with NJ also exhibit altered gut bile acid metabolism. In the present study, we found that gut bile acid levels (cholic acid) were significantly higher, and tauroursodeoxycholic acid levels were significantly lower in patients with NJ than in HCs, whereas tauroursodeoxycholic acid levels were negatively correlated with serum TBIL levels. These findings suggest that bile acid metabolic alterations may be a common feature of both NJ and ASD.

The gut DNA virome, comprising both eukaryotic viruses and bacteriophages, is now recognized as a key modulator of human health and disease through its interactions with the gut microbiota and host immunity (65, 66). Bacteriophages are viruses that infect intestinal bacteria and represent one of the most abundant members of the gut microbiome (67). There is an interaction between phages and bacteria. On one hand, phages can lyse bacteria, thereby regulating bacterial populations and maintaining gut microbiota balance (46, 47). On the other hand, symbiotic relationships exist between phages and bacteria, as exemplified by temperate phages integrating into bacterial genomes for co-evolution (47, 68). Although intrauterine infections such

as cytomegalovirus and hepatitis B virus are established risk factors for NJ—particularly cholestatic jaundice (69)—the role of the gut DNA virome in NJ remains poorly understood. In this study, we provided an initial characterization of the gut DNA virome in NJ, observing distinct viral community structures between NJ and HCs. Furthermore, both in the NJ and ASD cohorts, we observed a potential functional association between the gut DNA virome and bile acid metabolism genes. Based on these findings and supported by recent literature, we propose several mechanistic hypotheses that may explain how virome composition influences bile acid-metabolizing bacteria. First, shifts in phage abundance could directly modulate populations of bile acid-metabolizing bacteria through predator-prey interactions (70). Second, viral-encoded auxiliary metabolic genes (AMGs) may alter bacterial metabolic capacity (71), although such AMGs have not yet been fully characterized in bile acid pathways. Third, broader virome-mediated reshaping of the bacterial community (72) may have created niche opportunities that indirectly favor or suppress specific bile acid-modulating taxa. Finally, the gut DNA virome may trigger host immune responses that secondarily alter the gut environment and bacterial function (73).

The specific role of viruses in ASD is currently unknown. What is known is that administration of poly I:C, a molecule that mimics viral infection, to pregnant mice at day 12.5 of gestation can directly cause ASD-like behavior in their offspring, whereas the development of ASD-like behavior in offspring is dependent on the gut microbiota (2). A clinical FMT trial by Kang et al. reported concurrent alleviation of ASD symptoms and shifts in gut viral and bacterial communities following treatment (11). As observed for gut bile acid metabolism genes, a clear distinction could also be made between children with ASD and TD children based on the DNA viral composition, and notably, there was a potential concordance between gut bile acid metabolism genes and the ASD-specific DNA viral composition. In short, our results found a potential interaction between the gut DNA viral composition and gut bile acid metabolism genes.

Identifying important gut microbial, metabolite, viral, and phenotypic features from high-throughput multi-omic data, such as metagenomic, metabolomic, and viromic data, as well as host phenotypic features, is a challenging task. In the present study, in addition to gut microbiome–metabolome association analysis, we identified key bile acid metabolism genes and changes of the DNA viral composition affecting NJ and ASD by causal mediation analysis. Our results suggest that the gut DNA virome may influence NJ both directly and through bile acid metabolism genes, affecting core clinical indicators such as DBIL and TBIL. Similarly, in ASD, the virome may contribute to disease phenotype and bilirubin metabolism via bile acid-related pathways. Using machine learning-based causal inference, human betaherpesviruses and mastadenoviruses were implicated in the development of NJ and ASD, respectively, mediated by bile acid metabolic genes. These observations suggest that virome-induced shifts in bile acid composition might partially explain the link between NJ and ASD.

The gut microbiota plays a direct role in bile acid metabolism. As is known, cholesterol is generated in the liver by cholesterol 7-hydroxylase (CYP7A1) and sterol 27A hydroxylase (CYP27A1), discharged into the intestine through the gallbladder, hydrolyzed and desalted by bacterial BSH in the gut, and further transformed into secondary bile acids by oxidation under the action of gut bacterial HSDH (23). The majority (more than 95%) of bile acids in the gut can be reabsorbed by the liver through the enterohepatic circulation, and only a small amount (5%) of bile acids is excreted in feces (23). Structural and functional perturbations in the gut microbiome can disrupt this process. In our study, both NJ and ASD patients showed increased diversity of bile acid metabolism genes. Individuals with ASD also exhibited a higher abundance of bile acid-metabolizing bacteria (e.g., *Eggerthella*) and elevated IgA levels. Based on causal mediation and machine learning causal inference analyses, we found that bile acid metabolism was influenced by the composition of gut viruses such as human mastadenoviruses and human betaherpesviruses. In summary, we hypothesized that the altered abundance of bile acid-metabolizing bacteria caused by changes in the DNA viral

composition might play extremely important roles in the development of NJ and ASD and established a link between NJ and ASD (see Fig. 5). Currently, we have not yet fully identified all gut bacterial species capable of expressing BSH or HSDH. Our next step involves isolating these specific bacterial strains and conducting 16S rRNA sequencing for accurate identification.

A key aspect of this study is the use of a multi-omic framework, incorporating metagenomic, metabolomic, and viromic data to identify functional genes and viral components involved in bile acid metabolism in NJ and ASD. Furthermore, the combination of causal mediation and machine learning approaches helped infer potential virus-bile acid interactions underlying disease mechanisms.

However, this study had several limitations. First, the causal relationships identified here are derived from computational inference and observational data. While these models highlight plausible mechanistic pathways, future experimental validations using animal models or interventional studies are necessary to definitively establish causality. Second, the shotgun metagenomic libraries we analyzed were limited to DNA, meaning our findings provide a comprehensive view of the gut DNA virome but do not encompass the role of RNA viruses. Given the potential significance of RNA viruses in human health and disease, future investigations that incorporate RNA sequencing will be crucial to obtain a complete picture of the entire viral community and its interactions with the host and bacteriome. Furthermore, due to budgetary constraints, 16S rRNA sequencing was employed in NJ cohort 1, limiting the resolution of gene-level metabolic profiling. Additionally, the metabolome was not completed in NJ cohort 2 because of the lack of a sufficient number of completed macrogenomic sequencing samples. Furthermore, there are many variables that affect the gut microbiome, including host genetic factors, but these factors are difficult to control. Finally, our parent questionnaire showed that only 7 of 47 ASD children had a history of NJ. The sample size of the ASD cohort remains relatively small, and the retrospective collection of NJ history from parents is susceptible to recall bias, potentially leading to underreporting of milder cases. Future prospective studies with larger cohorts, systematic tracking of perinatal exposures, and functional experiments are needed to verify and extend these findings.

Currently, there are no universally effective treatments for ASD, although dietary interventions, probiotics, and FMT have shown promise in alleviating symptoms, possibly through microbial and metabolic restoration (74–76). Our analyses suggest that targeting the virome-microbiota-bile acid interface might offer a potential therapeutic strategy for ASD, a hypothesis that merits further investigation.

## Conclusion

Patients with NJ and ASD exhibit alterations of gut bile acid metabolism that are influenced by the DNA viral composition, and alterations in the DNA viral composition and bile acid-metabolizing bacteria appear to explain the link between NJ and ASD.

## ACKNOWLEDGMENTS

This study was financially supported by National Natural Science Foundation of China (No. 82571963), Natural Science Foundation of Guangdong Province, China (No.2025A1515012162, No.2024A1515010590), Shenzhen Science and Technology Program (No.s JCYJ20250604145739052 and JCYJ20240813144117023), the Research Initiation Fund of Longgang District Maternity and Child Healthcare Hospital of Shenzhen City (No. Y2024001), and by the Shenzhen Municipal Human Resources and Social Security Bureau (Postdoctoral fellow stationed in Shenzhen, second batch in 2022).

We would like to thank all participants for their invaluable contributions to this research.

X.C.: Data curation, investigation, formal analysis, visualization, writing—original draft, and writing—review and editing. C.C.: Data curation, investigation, formal analysis, visualization, and writing—original draft. X.L.: Data curation, investigation, formal analysis, visualization, and writing—original draft. X.Z.: Data curation, investigation, and

formal analysis. T.L.: Methodology and investigation. P.Z.: Methodology and investigation. G.C.: Investigation. W.Z.: Investigation. Z.W.: Investigation. Y.X.: Investigation. S.Z.: Supervision, project administration, and writing—review and editing. W.Z.: Conceptualization, supervision, project administration, resources, and writing—review and editing. M.W.: Conceptualization, supervision, funding acquisition, project administration, resources, and writing—review and editing.

## AUTHOR AFFILIATIONS

[1]Shenzhen Clinical Medical College, Guangzhou University of Chinese Medicine, Shenzhen, Guangdong, China

[2]Department of Neonatology, Affiliated Shenzhen Women and Children's Hospital (Longgang) of Shantou University Medical College (Longgang District Maternity and Child Healthcare Hospital of Shenzhen City), Shenzhen, Guangdong, China

[3]Division of Neonatology, Longgang Central Hospital of Shenzhen, Shenzhen, Guangdong, China

[4]Department of Geriatrics, Renji Hospital, Shanghai Jiao Tong University School of Medicine, Shanghai, China

[5]Division of Neonatology, Shenzhen Longhua People's Hospital, Shenzhen, Guangdong, China

[6]Neonatal Medical Center, Children's Hospital of Fudan University, Shanghai, China

[7]Division of Neonatology, Guangzhou Women and Children's Medical Center, Guangzhou Medical University, Guangzhou, Guangdong, China

## AUTHOR ORCIDs

Xianhong Chen  http://orcid.org/0009-0005-6805-4587
Shujuan Zeng  http://orcid.org/0000-0003-1095-0184
Wenhao Zhou  http://orcid.org/0000-0001-8956-7238
Mingbang Wang  http://orcid.org/0000-0002-5989-5377

## FUNDING

| Funder | Grant(s) | Author(s) |
| --- | --- | --- |
| National Natural Science Foundation of China | 82571963 | Mingbang Wang |
| Natural Science Foundation of Guangdong Province | 2025A1515012162, 2024A1515010590 | Mingbang Wang |
| Shenzhen Science and Technology Program | JCYJ20250604145739052 and JCYJ20240813144117023 | Mingbang Wang |
| Resear Initaion Fund of Longgang District Maternity & Child Healthcare Hospital of Shenzhen City | Y2024001 | Mingbang Wang |
| Shenzhen Municipal Human Resources and Social Security Bureau | Postdoctoral fellow stationed in Shenzhen second batch in 2022 | Mingbang Wang |

## AUTHOR CONTRIBUTIONS

Xianhong Chen, Data curation, Formal analysis, Investigation, Visualization, Writing – original draft, Writing – review and editing | Cheng Chen, Data curation, Formal analysis, Investigation, Visualization, Writing – original draft | Xiucai Lan, Data curation, Formal analysis, Investigation, Visualization, Writing – original draft | Xueli Zhang, Data curation, Formal analysis, Investigation | Tingting Li, Investigation, Methodology | Peng Zhang, Investigation, Methodology | Guoqiang Cheng, Investigation | Wei Zhou, Investigation | Zhangxing Wang, Investigation | Yingmei Xie, Investigation | Shujuan Zeng, Project administration, Supervision, Writing – review and editing | Wenhao Zhou,

Conceptualization, Project administration, Resources, Supervision, Writing – review and editing | Mingbang Wang, Conceptualization, Funding acquisition, Project administration, Resources, Supervision, Writing – review and editing

## DATA AVAILABILITY

The data of the NJ cohort 1 project have been uploaded to CNSA with project ID CNP0003066. The data of the ASD cohort and NJ cohort 2 projects have been uploaded to NCBI and can be downloaded according to the BioProject IDs, namely, PRJEB23052 and PRJEB23053.

## ETHICS APPROVAL

This study complied with the principles of the Declaration of Helsinki and was approved by the Medical Ethics Committees of Longgang District Maternity & Child Healthcare Hospital (Approval No. LGFYKYXMLL-2024-100). Written informed consent was obtained from the parents of each participant.

## ADDITIONAL FILES

The following material is available online.

### Supplemental Material

**Supplemental Material (mSystems01405-25-s0001.docx).** Supplemental figures and methods.
**Supplemental Tables (mSystems01405-25-s0002.xlsx).** Tables S1 to S8.

### Open Peer Review

**PEER REVIEW HISTORY (review-history.pdf).** An accounting of the reviewer comments and feedback.

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
