## [Reviewer comments · mSystems]

Machine learning and causal inference applied to the gut metagenome–metabolome axis reveals a link between neonatal jaundice and autism spectrum disorder

Xianhong Chen, Cheng Chen, Xiucai Lan, Xueli Zhang, Tingting Li, Peng Zhang, Guoqiang Cheng, Wei Zhou, Zhangxing Wang, Yingmei Xie, Shujuan Zeng, Wenhao Zhou, and Mingbang Wang

Corresponding Author(s): Mingbang Wang, Affiliated Shenzhen Women and Children's Hospital (Longgang) of Shantou University Medical College

Review Timeline:

Submission Date:	September 30, 2025
Editorial Decision:	November 6, 2025
Revision Received:	November 11, 2025
Accepted:	December 4, 2025

Editor: Hongwei Zhou

Reviewer(s): Disclosure of reviewer identity is with reference to reviewer comments included in decision letter(s). The following individuals involved in review of your submission have agreed to reveal their identity: Xingyin Liu (Reviewer #2)

Transaction Report:

DOI: <https://doi.org/10.1128/msystems.01405-25>

Re: mSystems01405-25 (**Machine learning and causal inference applied to the gut metagenome-metabolome axis reveals a link between neonatal jaundice and autism spectrum disorder**)

Dear Dr. Mingbang Wang:

Revision Guidelines

Sincerely,
Hongwei Zhou
Editor
mSystems

Reviewer #2 (Comments for the Author):

Major comments:

1. It proposes a common axis linking the gut virome, bile acid-metabolizing bacteria, and disease, connecting neonatal jaundice (NJ) and autism spectrum disorder (ASD). However, the two conditions are analyzed as separate case-control studies, without integrative statistics to assess whether microbial/viral signals are consistently altered in the same direction across cohorts.

Specifically, differential taxa/genes were identified independently (NJ vs HC, ASD vs TD) and were not compared side-by-side. Effect sizes for the top ASD-associated genera/BSH genes were not applied to the NJ dataset, and the proportion of features showing concordant directionality between NJ and ASD is not reported.

2. Please clarify that the current DNA metagenomic approach captures only DNA viruses. The terms 'gut virome' and 'viral composition' should be explicitly defined as 'DNA virome' or 'DNA viral reads' to reflect the study's scope. Furthermore, in the Discussion, should note that 'due to the DNA-only nature of the shotgun metagenomic libraries, the present findings are limited to DNA viruses. Future exploration of RNA viruses will be crucial...'.
3. To eliminate potential false positives from cross-mapping in PathoScope, it is recommended to select representative samples from the cohort and quantify viral copy numbers using ddPCR/qPCR.

4. The causal mediation pathways presented in Figure 3 lack any information on model stability. Please add a footnote providing details on the stability tests and results for these models.

5. Figure 3 currently lacks directional arrows. Adding arrows would clearly indicate which pathways are 'increasing' NJ or ASD risk. Alternatively, it should be stated that all causal pathways in the figure suggest an increase in risk, particularly with regard to the gut virome's influence on NJ, ASD, and other relevant levels through bile acid metabolism genes.

Reviewer comments:

Reviewer #2 (Comments for the Author):

Major comments:

1. It proposes a common axis linking the gut virome, bile acid-metabolizing bacteria, and disease, connecting neonatal jaundice (NJ) and autism spectrum disorder (ASD). However, the two conditions are analyzed as separate case-control studies, without integrative statistics to assess whether microbial/viral signals are consistently altered in the same direction across cohorts. Specifically, differential taxa/genes were identified independently (NJ vs HC, ASD vs TD) and were not compared side-by-side. Effect sizes for the top ASD-associated genera/BSH genes were not applied to the NJ dataset, and the proportion of features showing concordant directionality between NJ and ASD is not reported.

Response: Thank you for pointing out this issue. We identified consistently enriched or depleted gut markers, including microbiota, viruses, bile acid-metabolizing bacteria, differentially expressed genes, GBS genes, and metabolites across both the ASD and NJ cohorts. The results showed that 70 gut markers were significantly upregulated in the disease groups (ASD and NJ), consisting of 66 gut bacteria, 2 gut bile acid metabolism genes, 1 GBS gene, and 1 gut metabolite. Meanwhile, 33 gut markers were significantly downregulated in the disease groups, including 31 bacteria, 1 GBS gene, and 1 gut bile acid metabolism gene. Detailed results are provided in **Supplementary Table 8**. We have also revised the manuscript accordingly. (Manuscript Line 331-338)

2. Please clarify that the current DNA metagenomic approach captures only DNA viruses. The terms 'gut virome' and 'viral composition' should be explicitly defined as 'DNA virome' or 'DNA viral reads' to reflect the study's scope. Furthermore, in the Discussion, should note that 'due to the DNA-only nature of the shotgun metagenomic libraries, the present findings are limited to DNA viruses. Future exploration of RNA viruses will be crucial...'

Response: Thank you for raising this important point. We fully agree that explicitly stating the study's exclusive focus on DNA viruses is crucial for ensuring the rigor and clarity of our findings. Following your recommendation, we have implemented the following revisions throughout the manuscript.

(1) We have systematically updated terms such as "gut virome" and "viral composition" to the more precise "gut DNA virome" and "DNA viral composition" across the entire text, including the Abstract, Introduction, Methods, Results, and Discussion sections.

(2) In the Methods section under "Gut DNA virome," we have added the following statement: "It is important to note that the shotgun metagenomic sequencing approach employed in this study captures only DNA sequences. Consequently, our analysis of the viral community is confined to the DNA virome, and RNA viruses

were not detected." (Manuscript Line 177-180)

(3) We have expanded the Limitations paragraph in the Discussion section with this addition: "Secondly, the shotgun metagenomic libraries we analyzed were limited to DNA, meaning our findings provide a comprehensive view of the gut DNA virome but do not encompass the role of RNA viruses. Given the potential significance of RNA viruses in human health and disease, future investigations that incorporate RNA sequencing will be crucial to obtain a complete picture of the entire viral community and its interactions with the host and bacteriome." (Manuscript Line 466-471)

3. To eliminate potential false positives from cross-mapping in PathoScope, it is recommended to select representative samples from the cohort and quantify viral copy numbers using ddPCR/qPCR.

Response: Thank you for raising this point. We fully concur with the value of your suggestion. However, we sincerely regret to inform you that this study is a retrospective analysis based on historical cohorts from our previously published work. All of the precious original DNA samples were completely exhausted after a series of metagenomic sequencing procedures and preliminary experiments, making it currently impossible to obtain additional material for the suggested ddPCR/qPCR validation. We deeply apologize for this limitation and will ensure to reserve samples for such validation assays in our future studies.

4. The causal mediation pathways presented in Figure 3 lack any information on model stability. Please add a footnote providing details on the stability tests and results for these models.

Response: Thank you for pointing out this issue. The stability of the model was assessed by incorporating a bootstrap procedure into the causal mediation analysis, with the number of bootstrap replications set to 999. This process yielded the point estimates (Estimate), 95% confidence intervals, and p-values for the ACME, ADE, and Total Effect. The results indicate that a p-value for the ACME of less than 0.05 signifies the presence of a stable causal mediation effect. (Manuscript Line 801-803)

5. Figure 3 currently lacks directional arrows. Adding arrows would clearly indicate which pathways are 'increasing' NJ or ASD risk. Alternatively, it should be stated that all causal pathways in the figure suggest an increase in risk, particularly with regard to the gut virome's influence on NJ, ASD, and other relevant levels through bile acid metabolism genes.

Response: Thank you for your valuable feedback. In response to your suggestion regarding the lack of directional arrows in Figure 3 to indicate the path directions, we fully agree. To more clearly illustrate how different pathways influence NJ or ASD risk, we have added directional arrows to each subplot in Figure 3. As you noted, all causal pathways in the figure indicate an increased risk. We have also included a note in the figure caption clarifying that upward-pointing arrows represent an increase in

disease risk or an elevation in measured indicators. (Manuscript Line 803-804)

Re: mSystems01405-25R1 (**Machine learning and causal inference applied to the gut metagenome-metabolome axis reveals a link between neonatal jaundice and autism spectrum disorder**)

Dear Dr. Mingbang Wang:

Your manuscript has been accepted, and I am forwarding it to the ASM production staff for publication. Your paper will first be checked to make sure all elements meet the technical requirements. ASM staff will contact you if anything needs to be revised before copyediting and production can begin. Otherwise, you will be notified when your proofs are ready to be viewed.

Sincerely,
Hongwei Zhou
Editor
mSystems